# On Positive Definite Kernels of Integral Operators Corresponding to the Boundary Value Problems for Fractional Differential Equations

**DOI:** 10.3390/e24040515

**Published:** 2022-04-06

**Authors:** Mukhamed Aleroev, Temirkhan Aleroev

**Affiliations:** 1Department of Higher Mathematics, Moscow Automobile and Road Construction State Technical University (STU-MADI), Leningradsky Ave. 64, 125319 Moscow, Russia; mahammad.aleroev@yandex.ru; 2Department of Applied Mathematics, The National Research Moscow State University of Civil Engineering (NRU MGSU), Yaroslavskoe Highway 26, 129337 Moscow, Russia

**Keywords:** persymmetric matrix, eigenvalues, fractional derivative, positive definiteness

## Abstract

In the spectral analysis of operators associated with Sturm–Liouville-type boundary value problems for fractional differential equations, the problem of positive definiteness or the problem of Hermitian nonnegativity of the corresponding kernels plays an important role. The present paper is mainly devoted to this problem. It should be noted that the operators under study are non-self-adjoint, their spectral structure is not well investigated. In this paper we use various methods to prove the Hermitian non-negativity of the studied kernels; in particular, a study of matrices that approximate the Green’s function of the boundary value problem for a differential equation of fractional order is carried out. Using the well-known Livshits theorem, it is shown that the system of eigenfunctions of considered operator is complete in the space $L_{2}\left(0,1\right)$. Generally speaking, it should be noted that this very important problem turned out to be very difficult.

## 1. Introduction

In the theory of oscillations, the general physical process of reciprocity (when the source and the observer can exchange places) finds its mathematical expression in the self-adjointness of the corresponding boundary value problems [1,2]. When we talk about processes corresponding to structures with fractal geometry, the corresponding processes are described by non-self-adjoint operators, and here, first of all, we are talking about integral operators with a persymmetric kernel [1,2]. The spectral structure of such operators has hardly been studied. The present manuscript is devoted to this question.

## 2. Main Results

In the present paper we consider the following operator
$$-A_{\rho }u=\frac{1}{\Gamma (\rho^{-1})}\left\{\int_{0}^{x}{\left(x-t\right)}^{\frac{1}{\rho }-1}u\left(t\right)dt-\int_{0}^{1}{\left(x-xt\right)}^{\frac{1}{\rho }-1}u\left(t\right)dt\right\},\;\left(1/2<\rho <1\right)$$
Corresponding to the following problem [3]
$$D^{1/\rho }u=\frac{1}{\Gamma (\rho^{-1})}\frac{d}{dx}\int_{0}^{x}\frac{u^{\prime }\left(t\right)dt}{{\left(x-t\right)}^{1/\rho -1}}=\lambda u,$$
u(0)=u(1)=0.

By the Livshits theorem [2,4], the system of eigenfunctions of this operator is complete in *L*${}_{2}$(0, 1); precisely, we have

**Theorem** **1.**
* (Livshits): if K(x,y),(a≤x,y≤b) is a bounded kernel, and the “real part” $\frac{1}{2}\left(K+K^{*}\right)$ of it is non-negative kernel, then the inequality holds*

$$\sum_{j=1}^{\infty }Re\left(1/\lambda_{j}\right)\leq \int_{a}^{b}ReK\left(t,t\right)dt,$$

*where $\lambda_{j}$ is the characteristic numbers of kernel K. The system of main functions of the kernel K is complete in the domain of values of the integral operator Kf if and only if, when there is an equal sign in the inequality above.*


Here, in this theorem characteristic numbers are eigenvalues. It is known that the operator *A* is called positive definite (the definition of positivity of the operator and its properties can be found in [5]) if (Au,u)>0,(u≠0). However, it is very difficult to verify this condition directly. Therefore, we will use the matrix approximation of the operator $A_{\rho }$ [1]. As in [1], we denote the corresponding matrix by $T_{n-1}\left(\mu \right),\;\mu =\frac{1}{\rho }-1$
$$T_{n-1}\left(\mu \right)=\left(\begin{array}{cccc} {\left(\frac{1}{n}\right)}^{\mu }{\left(\frac{n-1}{n}\right)}^{\mu } & {\left(\frac{1}{n}\right)}^{\mu }{\left(\frac{n-2}{n}\right)}^{\mu } & \cdots & {\left(\frac{1}{n}\right)}^{\mu }{\left(\frac{1}{n}\right)}^{\mu }\\ {\left(\frac{2}{n}\right)}^{\mu }{\left(\frac{n-1}{n}\right)}^{\mu }-{\left(\frac{1}{n}\right)}^{\mu } & {\left(\frac{2}{n}\right)}^{\mu }{\left(\frac{n-2}{n}\right)}^{\mu } & \cdots & {\left(\frac{2}{n}\right)}^{\mu }{\left(\frac{1}{n}\right)}^{\mu }\\ \vdots & \vdots & \ddots & \vdots \\ {\left(\frac{n-1}{n}\right)}^{\mu }{\left(\frac{n-1}{n}\right)}^{\mu }-{\left(\frac{n-1}{n}\right)}^{\mu } & {\left(\frac{n-1}{n}\right)}^{\mu }{\left(\frac{n-2}{n}\right)}^{\mu }-{\left(\frac{n-3}{n}\right)}^{\mu } & \cdots & {\left(\frac{n-1}{n}\right)}^{\mu }{\left(\frac{1}{n}\right)}^{\mu } \end{array}\right).$$

The matrix $T_{n-1}\left(\mu \right)$ has many useful properties. In particular, this matrix is positive, persymmetric, indecomposable, etc. It is known [1] that one of the necessary conditions for the positive definiteness of a matrix is the positivity of all its lead main minors. The fact that these minors are positive was shown in [1,6]. We give a detailed proof of one theorem from which the above follows.

**Theorem** **2.**
*The minors*

$$A\left(\begin{array}{cccc} i_{1} & i_{2} & \cdots & i_{r}\\ j_{1} & j_{2} & \cdots & j_{r} \end{array}\right)$$

*of the matrix*

$$T_{n-1}\left(\mu \right)=\left(\begin{array}{cccc} {\left(\frac{1}{n}\right)}^{\mu }{\left(\frac{n-1}{n}\right)}^{\mu } & {\left(\frac{1}{n}\right)}^{\mu }{\left(\frac{n-2}{n}\right)}^{\mu } & \cdots & {\left(\frac{1}{n}\right)}^{\mu }{\left(\frac{1}{n}\right)}^{\mu }\\ {\left(\frac{2}{n}\right)}^{\mu }{\left(\frac{n-1}{n}\right)}^{\mu }-{\left(\frac{1}{n}\right)}^{\mu } & {\left(\frac{2}{n}\right)}^{\mu }{\left(\frac{n-2}{n}\right)}^{\mu } & \cdots & {\left(\frac{2}{n}\right)}^{\mu }{\left(\frac{1}{n}\right)}^{\mu }\\ \vdots & \vdots & \ddots & \vdots \\ {\left(\frac{n-1}{n}\right)}^{\mu }{\left(\frac{n-1}{n}\right)}^{\mu }-{\left(\frac{n-1}{n}\right)}^{\mu } & {\left(\frac{n-1}{n}\right)}^{\mu }{\left(\frac{n-2}{n}\right)}^{\mu }-{\left(\frac{n-3}{n}\right)}^{\mu } & \cdots & {\left(\frac{n-1}{n}\right)}^{\mu }{\left(\frac{1}{n}\right)}^{\mu } \end{array}\right).$$

*for $i_{k}\leq j_{k},\;1\leq k\leq r,$ are positive. Moreover, they are equal to*

$${\left(n^{\mu }\right)}^{r-1}{\left(n-j_{r}\right)}^{\mu }{\left(i_{1}\right)}^{\mu }b_{r(r-1)}b_{(r-1)(r-2)\ldots }b_{21},$$


*where*

$$b_{ki}=\left\{\begin{array}{c} {\left(i_{k}-j_{i}\right)}^{\mu },\;i_{k}>j_{i}\\ 0,\;i_{k}<j_{i} \end{array}\right.$$



**Proof.** Let us consider the minor
$$M_{r}=A\left(\begin{array}{cccc} i_{1} & i_{2} & \cdots & i_{r}\\ j_{1} & j_{2} & \cdots & j_{r} \end{array}\right).$$For $i_{k}\leq j_{k},\;1\leq k\geq r,$ we to overwrite $M_{r}$ as follows
$$M_{r}=\left(\begin{array}{c} i_{1}^{\mu }\\ i_{2}^{\mu }\\ i_{3}^{\mu }\\ \vdots \\ i_{r1}^{\mu } \end{array}\right)\left({\left(n-j_{1}\right)}^{\mu }{\left(n-j_{2}\right)}^{mu}\ldots {\left(n-j_{r}\right)}^{\mu }\right)-n^{\mu }\left(\begin{array}{ccccc} 0 & 0 & 0 & \cdots & 0\\ b_{21} & 0 & 0 & \cdots & 0\\ \vdots & \vdots & \ddots & \vdots \\ b_{r1} & b_{r2} & b_{r3} & \cdots & b_{r(n-1)} \end{array}\right)$$for
$$b_{ki}=\left\{\begin{array}{c} {\left(i_{k}-j_{i}\right)}^{\mu },\;i_{k}>j_{i}\\ 0,\;i_{k}<j_{i} \end{array}\right.$$To calculate the determinant, $M_{r}$ we consider
$$\det\left(M_{r}-\lambda I\right)={\left(-1\right)}^{r}\det\left(\lambda I-M_{r}\right)={\left(-1\right)}^{r}\det\left(\tilde{A}-xy^{T}\right)=\;{\left(-1\right)}^{r}\lambda^{r}\left(1-y^{T}{\tilde{A}}^{-1}x\right).$$Here,
$$\tilde{A}=n^{\mu }\left(\begin{array}{ccccc} \frac{\lambda }{n^{\mu }} & 0 & 0 & \cdots & 0\\ b_{21} & \frac{\lambda }{n^{\mu }} & 0 & \cdots & 0\\ \vdots & \vdots & \ddots & \vdots \\ b_{r1} & b_{r2} & b_{r3} & \cdots & \frac{\lambda }{n^{\mu }} \end{array}\right),$$
$$x=\left(\begin{array}{c} i_{1}^{\mu }\\ i_{2}^{\mu }\\ i_{3}^{\mu }\\ \vdots \\ i_{r1}^{\mu } \end{array}\right),y^{T}=\left({\left(n-j_{1}\right)}^{\mu }{\left(n-j_{2}\right)}^{\mu }\ldots {\left(n-j_{r}\right)}^{\mu }\right).$$It is clear that
$$x_{r1}={\left(-\frac{n^{\mu }}{\lambda }\right)}^{r-1}b_{r(r-1)}b_{(r-1)(r-2)\ldots }b_{21}x_{1}\ldots =\;={\left(-1\right)}^{r-1}{\left(\frac{n^{\mu }}{\lambda }\right)}^{r-1}b_{r(r-1)}b_{(r-1)(r-2)\ldots }b_{21}+\ldots$$So,
$$\det\left(M_{r}-\lambda I\right)={\left(-1\right)}^{r}\lambda^{r}\left(1-y^{T}{\tilde{A}}^{-1}x\right)=\;{\left(-1\right)}^{r}\lambda_{r}\left(1-{\left(n-j_{r}\right)}^{\mu }i_{1}^{r}x_{r1}+\ldots \right)$$from this follows
$$\det\left(M_{r}\right)={\left(n^{\mu }\right)}^{r-1}{\left(n-j_{r}\right)}^{\mu }{\left(i_{1}\right)}^{\mu }b_{r(r-1)}b_{(r-1)(r-2)\ldots }b_{21}$$
that proves Theorem 2.  □

To prove that matrix $T_{n-1}\left(\mu \right)$ is positive defined, we have
$$T_{R}=\frac{1}{2}\left(T_{n-1}\left(\mu \right)+T_{n-1}^{*}\left(\mu \right)\right).$$

It is obvious that the matrix $T_{R}$, in addition to everything else, is also bisymmetric (symmetric with respect to both the main and secondary diagonals). Using the high-level mathematical package MATLAB, the eigenvalues of the matrix $T_{R}$ were considered for various values of μ and the dimension of the matrix *N*. It was shown that all eigenvalues of the matrix $T_{R}$, for any N≤3000 and μ>0, are positive; that is, the above calculations confirm the hypothesis that the matrix $T_{R}$ is positive definite. This became the basis for us to assume that the matrix $T_{\left(n-1\right)}\left(\mu \right)$ under study is positive definite. Naturally, the operator $A_{\rho }$ corresponding to the matrix $T_{\left(n-1\right)}\left(\mu \right)$, will also be positive definite [7,8].

We give a strong proof of the positive definiteness of the matrix $T_{R}\left(\mu \right)$. First, let us write out the matrices $T_{6}\left(1/2\right)$, $T_{6}^{*}\left(1/2\right)$, $T_{R}\left(1/2\right)$ using the MATLAB package
$$T_{6}\left(1/2\right)=\left(\begin{array}{cccccc} 2.4495 & 2.2361 & 2 & 1.7321 & 1.4142 & 1\\ 0.8184 & 3.1623 & 2.8284 & 2.4495 & 2 & 1.4142\\ 0.5010 & 1.2272 & 3.4641 & 3 & 2.4495 & 1.7321\\ 0.3164 & 0.7305 & 1.3542 & 3.4641 & 2.8284 & 2\\ 0.1857 & 0.4174 & 0.7305 & 1.2272 & 3.1623 & 2.2361\\ 0.0839 & 0.1857 & 0.3164 & 0.5010 & 0.8184 & 2.4495 \end{array}\right)$$
$$T_{6}^{*}\left(1/2\right)=\left(\begin{array}{cccccc} 2.4495 & 0.8184 & 0.5010 & 0.3164 & 0.1857 & 0.0839\\ 2.2361 & 3.1623 & 1.2272 & 0.7305 & 0.4174 & 0.1857\\ 2 & 2.8284 & 3.4641 & 1.3542 & 0.7305 & 0.3164\\ 1.7321 & 2.4495 & 3 & 3.4641 & 1.2272 & 0.5010\\ 1.4142 & 2 & 2.4495 & 2.8284 & 3.1623 & 0.8184\\ 1 & 1.4142 & 1.7321 & 2 & 2.2361 & 2.4495 \end{array}\right)$$
$$T_{R}\left(1/2\right)=\left(\begin{array}{cccccc} 2.4495 & 1.5272 & 1.2505 & 1.0242 & 0.8 & 0.5420\\ 1.5272 & 3.1623 & 2.0278 & 1.5900 & 1.2087 & 0.8\\ 1.2505 & 2.0278 & 3.4641 & 2.1771 & 1.59 & 1.0242\\ 1.0242 & 1.5900 & 2.1771 & 3.4641 & 2.0278 & 1.2505\\ 0.8 & 1.2087 & 1.59 & 2.0278 & 3.1623 & 1.5272\\ 0.5420 & 0.8 & 1.0242 & 1.5272 & 1.5272 & 2.4495 \end{array}\right)$$
A simple analysis of these matrices shows that the elements from the main diagonal (including the diagonal itself) increase in rows and columns from the edges to the main diagonal. That is, the following statements hold:

**Lemma** **1.**
*For any fixed $i_{0}\leq j$, the relations*

$$a_{i_{0},j}\geq a_{i_{0},j+1},\;i_{0}\leq j;$$


$$a_{i_{0},j}<a_{i_{0},j+1},\;i_{0}>j.$$



**Proof.** We write the formula for the general element of the matrix
$$a_{ij}={\left(Ni-ij\right)}^{\mu }-\theta \left(i,j\right){\left(Ni-Nj\right)}^{\mu },$$
where
$$\theta \left(i,j\right)=\left\{\begin{array}{c} 0,\;j\geq \;i\\ 1,\;j<i. \end{array}\right.$$
Obviously, the elements under the main diagonal are calculated as follows
$$a_{ij}={\left(Ni-ij\right)}^{\mu }-{\left(Ni-Nj\right)}^{\mu },\;i>j,$$
and the elements under the main diagonal are
$$a_{ij}={\left(Ni-ij\right)}^{\mu },\;i>j.$$
From these formulas, it follows that the elements located above the main diagonal decrease. To consider the elements under the main diagonal, we introduce the generating function
$$\varphi \left(x\right)={\left(Ni-ix\right)}^{\mu }-\theta \left(i,j\right){\left(Ni-Nx\right)}^{\mu },\;\mu \in \left(0,1\right),\;x\in \left[1,N\right].$$
Obviously, the derivative of this function is positive on the segment x∈[1,N], which means that the function φ(x) increases on the segment x∈[1,N]. That is the prove.    □

**Lemma** **2.**
*For any fixed $j_{0}<i$, the relations*

$$a_{i,j_{0}}\geq a_{i+1,j_{0}},\;i\geq j_{0};$$


$$a_{i,j_{0}}<a_{i+1,j_{0}},\;i<j_{0}$$

*hold.*


**Proof.** The proof of Lemma 2 is similar to the proof of Lemma 1.    □

**Lemma** **3.**
*The statements of Lemmas 1 and 2 are valid for the matrices $T_{n}^{T}\left(\mu \right)$ ($T_{n}^{T}\left(\mu \right)$ is the transposed matrix).*


**Lemma** **4.**
*The statements of Lemmas 1,2,3 are also valid for the matrices $T_{R}\left(\mu \right)=\frac{T_{n}\left(\mu \right)+T_{n}^{T}\left(\mu \right)}{2}$.*
Using these lemmas, we prove the following theorem

**Theorem** **3.**
*The matrix $T_{R}\left(\mu \right)=\frac{T_{n}\left(\mu \right)+T_{n}^{T}\left(\mu \right)}{2}$ is a positive defined for μ∈[0,1].*


**Proof.** It is obvious that all the main lead minors of the matrix $T_{R}\left(0\right)$ are non-negative. In the same way, all main lead minors of the matrix $T_{R}\left(1\right)$ are positive.Let us show for μ∈(0,1) that all main lead minors of the matrix are $T_{R}\left(\mu \right)\neq 0$. To do this, it is enough to prove that all the rows (columns) of the leading main lead minors of the matrix are linearly independent. In proving this statement, without loss of generality, for definiteness, we consider rows with numbers *k* and k+1. Then, it suffices to note that, by Lemma 4, $\frac{a_{k,1}}{a_{k+1,1}}<1$ and $\frac{a_{k,k+1}}{a_{k,k}}>1$, which proves the linear independence of these rows.Let us introduce the following function
$$detT_{R}\left(\mu \right)=\Delta \left(\mu \right),\;\mu \in \left[0,1\right].$$
It is known that Δ(0)≥0 and Δ(1)≥0.From the Theorem 3 it follows that the operator $A_{\rho }$ is positive definite for 1/2<ρ<∞.The proof of the positive definiteness of the operator $A_{\rho }$ for 1/2<ρ<∞ can also be carried out as follows. Let us define
$$\frac{1}{\Gamma (\frac{1}{\rho })}\left[\int_{0}^{x}{\left(x-t\right)}^{\frac{1}{\rho }-1}u\left(t\right)dt-\int_{0}^{1}x^{\frac{1}{\rho }-1}{\left(1-t\right)}^{\frac{1}{\rho }-1}u\left(t\right)dt\right]=v\left(x\right).$$We act on both sides of this equation by the operator $D^{\frac{1}{\rho }}$, where $D^{\frac{1}{\rho }}$ is the fractional differentiation operator in the Riemann–Liouville sense, then $u\left(x\right)=D^{\frac{1}{\rho }}v$ [5]. Then we may show that the form $(A_{\rho }u,u)>0.$In reality,
$$-\left(A_{\rho }u,u\right)=-\left(v,D^{\frac{1}{\rho }}v\right)=-\left(\frac{1}{\Gamma (\frac{1}{\rho })}\frac{d}{dx}\int_{0}^{x}\frac{f^{\prime }\left(t\right)}{{\left(x-t\right)}^{\frac{1}{\rho }-1}}dt,\;f\left(x\right)\right)=\;\left(\frac{1}{\Gamma (\frac{1}{\rho })}\frac{d}{dx}\int_{0}^{x}\frac{f^{\prime }\left(t\right)}{{\left(x-t\right)}^{\frac{1}{\rho }-1}}dt,\;f^{\prime }\left(x\right)\right)=\;(J_{\frac{1}{\rho }}f^{\prime },\;f^{\prime }),$$
where
$$\left(J_{\frac{1}{\rho }}f\right)\left(t\right)=\frac{1}{\Gamma (\frac{1}{\rho })}\int_{0}^{t}{\left(t-s\right)}^{1-\frac{1}{\rho }}f\left(s\right)ds$$
-is the operator of fractional integration in the Riemann–Liouville sense of order $\frac{1}{\rho }.$    □

Since [1] $(J_{\frac{1}{\rho }}f^{\prime },\;f^{\prime })>0,$ for $\frac{1}{\rho }<1$, and taking into account that operator $A_{\rho }$ is kernel [5,9], we prove the following theorem

**Theorem** **4.**
*The system of eigenfunctions and associated functions $A_{\rho }$ for $\frac{1}{2}<1/\rho <1$ is complete in $L_{2}\left(0,1\right).$*


**Corollary** **1.**
*Since the operator $A_{\rho }$ is positive definite, then all matrices $T_{\left(n-1\right)}\left(\mu \right)$ for n>N are positive definite.*


**Remark** **1.**
*The matrix $T_{\left(n-1\right)}\left(\mu \right)$ may be presented as*

$$T_{\left(n-1\right)}\left(\mu \right)=B^{*}B$$

*where B is the triangular matrix.*


## 3. Discussion

Operators generated (induced) by a differential expression of a fractional order and boundary conditions of the Sturm–Liouville type are non-self-adjoint and their spectral structure is almost not studied. The methods proposed by the authors are fundamentally new. They allow study of the completeness of systems of eigenfunctions and associated functions of these operators.

## 4. Conclusions

Thus, our spectral analysis of the operators generated by boundary value problems for fractional differential equations and boundary conditions of the Sturm–Liouville type, using matrix calculus, shows that the spectral structure of these operators can be studied by the matrices we studied above.

## Data Availability

Not applicable.

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
