# Peer review of "On Positive Definite Kernels of Integral Operators Corresponding to the Boundary Value Problems for Fractional Differential Equations"

_entropy, 2022, doi:10.3390/e24040515_

Round 1

Reviewer 1 Report

See enclosed report

Author Response

Dear reviewers! Thanks a lot for your useful remarks! Of course, we corrected all our mistakes. More attention I’d like to pay for comments of one reviewer, where was noticed that the references are not enough, but it connected with such fact that spectral structure of considered operators in our paper is not investigated completely, and these operators are non-self-adjoint. The spectral structure of operators is very difficult and certainly in future we will to design methods for studying this structure. Despite of very significant role of these operators in fractional calculus, there no any complete results.  

Reviewer 2 Report

Manuscript ID: ID AME-22-0176

Manuscript Title: On positive definite kernels of integral operators corresponding to the boundary value problems for fractional differential equations

Authors:  Mukhamed Aleroev  and Temirkhan Aleroev

Dear    Editor,

I have attached below the report.

Best wishes,

The authors studied the spectral analysis of operators associated with Sturm-Liouville-type boundary value  problems for fractional differential equations

They used various  methods to prove the Hermitian non-negativity of the studied kernels. They utilized the well-known Livshits theorem to shown that the system of eigenfunctions of considered operator is complete in the space L_{2}(0, 1)}.

I have some comments and questions that the authors should to do it, so that the paper will be looks better:

  • Explain your original contribution.
  • Proofreading review is required for the paper.
  • Discuss the advantages of the suggested methods over other existing methods.
  • Write motivation for the carried research work in introduction.
  • The English structure of the article, including punctuation, semicolon, and other structures, must be carefully reviewed.
  • The paper is not formatted in accordance with the journal requirement.
  • Try to avoid include the references and letters in the abstract.

After revisions I recommend y this paper for publication in entropy.

Author Response

(The authors gave the same response as above.)

Reviewer 3 Report

I believe that the topic of the study is important, the results obtained are interesting, but the presentation of the manuscript should be significantly revised.

I have the following comments. 

1. In the introduction, the history of the subject and previously obtained results are not described in sufficient detail.

2. Line 22. It is not very clear what the operator $A_\rho$ has to do with the boundary value problem below. What is the operator $D$ and how is its degree determined. What functions does it apply to and what is $\Gamma$ and $\lambda$. 

3. Line 23, "Using the well-known Livshits theorem [7]". It's not good to write "well-known theorem" referring to your own work, which can hardly be called "well-known". 

4. Theorem 1. Instead of "limited" it is better to write "bounded". 

5. Line 28. Which operator $A$ are you talking about and what is $(Au,u)$.

6. Line 30. It would be nice to explain the words "matrix approximation of the operator $A_\rho$". What approximation are you talking about?

7. There is no need to write the matrix $T_{n−1}(\mu)$ twice.

8. Line 39. Notation confusion: $i_k$ and $j_i$. It is better to replace the subscript in the expression $j_i$. 

9. Line 42. Since "a minor of a matrix is the determinant of some smaller square matrix", it is necessary to explain the equality in which the matrix is on the right-hand side and the minor $M_r$ is on the left-hand side.

10. Why does line 49 say "the real component of this matrix". The matrix $T_{n−1}(\mu)$ is a real matrix.

11. Line 54. The dimension of the square matrix $T_R$ is $n-1$. What does the index $R$ mean for this matrix and why is it necessary to introduce a new value $N$. 

12. The formulation of Lemma 1 is not completed. What are θ(i,j) and N? In the proof of Lemma 1, two sentences are the same.

13. Line 83. "From this very important theorem". What theorem are you talking about here? Where the proof of Theorem 3 ends.

14. Line 86. It is not clear where the equality $u(x)=D^{\frac1p}v$ follows from? Below, it is necessary to explain how the dot product is defined.

15. The proof of the main Theorem 4 is not clear and contains only references. How does Theorem 4 follow from Theorem 3? 

16. For the readers of the journal in Conclusions it would be good to discuss possible applications of the results obtained, as well as their further development. What are the possible ways to continue the research topic?

17. Of the 7 cited papers, 5 belong to the authors of the manuscript. This may indicate that the research topic of the article is very narrow and will not attract much attention from readers. 

Author Response

(The authors gave the same response as above.)

Reviewer 4 Report

My observations are:

  1. Delete the references from the abstract section.
  2. The introduction section is very short and can't cover the significant literature.
  3. Add references to the data used from the literature.
  4. Add applications and examples in order to justify the theorems and show who these theorems are good from the work that exist in literature. 

Author Response

(The authors gave the same response as above.)

Round 2

Reviewer 3 Report

Many of the remarks in the manuscript have been corrected, but a few remain unexplained.
1. It is necessary to indicate explicitly where the proof of Theorem 3 ends and the proof of Theorem 4 begins.
2. Line 90. It is written that $\frac1\rho<1$, and line 92 says the opposite $\rho<1$.
3. Where is Theorem 1 specifically applied in the proof of Theorem 4. 

Author Response

Thank you very much for our comments. All remarks were corrected. 

Reviewer 4 Report

I recommended for publication of this revised form. 

Author Response

Thank you very much for your review